# Development and validation of a machine learning-based predictive model for clinical remission in Crohn's disease patients receiving Adalimumab therapy

**Pianpian Xia**[1,2,3], **Feihong Deng**[1,2,3]*, **Deliang Liu**[1,2,3]

**1** Department of Gastroenterology, The Second Xiangya Hospital of Central South University, Changsha, Hunan, China, **2** Research Center of Digestive Diseases, Central South University, Changsha, Hunan, China, **3** Clinical Research Center for Digestive Diseases in Hunan Province, Changsha, Hunan, China

* dfh411@csu.edu.cn

## Abstract

Crohn's disease (CD), a chronic inflammatory bowel disease, is witnessing a rising global incidence. Adalimumab (ADA), a biological agent, is widely used in its treatment. However, patients exhibit significant individual variability in responses to ADA therapy. This study focuses on developing and validating a machine learning – based predictive model to assess the clinical remission of CD patients at 12 and 48 weeks post – ADA treatment, while identifying the key influencing factors. A single – center retrospective study was conducted, involving patients from the Second Xiangya Hospital of Central South University between 2017 and 2024. Comprehensive data on demographics, lifestyle, disease characteristics, and laboratory indicators were collected and preprocessed. The dataset was partitioned into an 80% training set and a 20% test set. Six machine learning models, including Random Forest and Gradient Boosting Machine, were employed to construct the prediction model. Model performance was evaluated using metrics such as accuracy, sensitivity, and specificity. The SHAP analysis was performed to elucidate the key factors. The results indicated that the XGBoost model outperformed other models across multiple evaluation metrics. Fecal calprotectin (Fc), a marker of intestinal inflammation, showed that lower levels were associated with a tendency towards mucosal healing. C - reactive protein (CRP), on the other hand, reflected systemic inflammation. Both biomarkers significantly influenced the prediction outcomes at different time points. The developed model serves as a valuable tool for clinical stratification and personalized treatment planning. Future research should expand sample diversity through multi – center collaboration and integrate multi – omics data, such as gut microbiome and metabolomics, to further enhance the model's ability to capture the molecular mechanisms underlying the disease.

**Data availability statement:** All relevant data are within the paper and its Supporting Information files.

**Funding:** Funding This work was supported by the National Natural Science Foundation of China (Grant number 82470586), and the grantee is Dr. Feihong Deng.

**Competing interests:** The authors have declared that no competing interests exist.

## Introduction

Crohn's disease (CD) is a chronic and recurrent inflammatory bowel disease, with its incidence exhibiting a significant upward trend globally [1,2]. Epidemiological data indicate that the incidence of CD in Europe and America has surpassed 20 per 100,000, while the annual growth rate in Asia ranges from 4% to 15%, posing a substantial threat to public health [3–5]. This disease can affect the entire thickness of the digestive tract, manifesting primarily as abdominal pain, diarrhea, weight loss, and extraintestinal complications, significantly impacting patients' quality of life [6]. Although anti-TNF-α drugs are the first-line therapy for moderate to severe CD, approximately 30%−40% of patients experience primary or secondary non-response. This challenge not only leads to treatment failure but also exacerbates the waste of medical resources [7,8].

Traditional clinical prediction models often rely on single indicators to assess the efficacy of adalimumab (ADA), yet the heterogeneity of Crohn's disease (CD) makes it challenging to fully capture the complexity of the condition. Recent advancements in machine learning technology have introduced novel approaches for precise and personalized outcome prediction. Although machine learning applications in predicting anti-TNF treatment responses for CD have demonstrated notable progress, they remain constrained by several limitations.

While prior studies, such as Qiu et al. (2024), concentrated on single algorithms for infliximab response prediction, they often lacked systematic comparisons across multiple models [9]. For instance, Jiang et al. (2023) applied deep learning to predict ulcerative colitis remission but failed to implement rigorous cross-validation, which may have led to an overestimation of model performance [10]. Furthermore, existing models seldom incorporate longitudinal data; Gradel et al.'s 2024 infliximab study, for example, only analyzed short-term (12-week) outcomes without considering long-term follow-up. Additionally, few studies account for dynamic changes in predictive factors over time [11]. Kawashima et al.'s 2023 study used SHAP analysis to explore static feature importance in ulcerative colitis but did not address the temporal evolution of these features [12].

This study aims to develop and validate a machine learning model to predict the clinical remission status of CD patients at 12 weeks and 48 weeks post – ADA treatment respectively, and to analyze key influencing factors using the SHAP algorithm. Compared with existing studies, this research systematically compares the predictive performance of six mainstream machine learning models and employs the SHAP algorithm to interpret feature contributions, providing interpretable support for clinical decision-making.

## Materials and methods

### Study design

This study employed a single-center retrospective design, including patients diagnosed with CD and treated with ADA at the Department of Gastroenterology, Second Xiangya Hospital of Central South University, from January 2017 to April 2024.

Inclusion criteria were: (1) age between 18 and 70 years; (2) confirmed diagnosis of CD according to the World Health Organization (WHO) criteria recommended by the World Gastroenterology Organisation (WGO); (3) follow-up period exceeding 12 months; (4) no prior use of non-TNF-α antagonist biological agents; (5) no CD-related abdominal or peri-anal surgery within one year before medication; and (6) complete clinical data. Although this was a retrospective study, informed consent was obtained from all participants, and this study was approved by the Ethics Committee of the Second Xiangya Hospital of Central South University (Number: LYF2023123).

## Data collection

Comprehensive patient information was collected, including demographic data (gender, age group based on Montreal classification), lifestyle information (smoking history, smoking amount, and duration), disease characteristics (disease behavior and anatomical location), disease location (small intestine, colon, ileocolon, etc.), and laboratory indicators before treatment. The laboratory indicators specifically refer to Body Mass Index (BMI), White Blood Cell Count (WBC), Hemoglobin (Hb), Platelet Count (plt), Albumin (Alb), C - reactive Protein (CRP), Erythrocyte Sedimentation Rate (ESR), and Fecal Calprotectin (Fc). Clinical remission was defined as a CD Activity Index (CDAI) score less than 150, in accordance with internationally recognized standards [13].

The Montreal classification is a widely adopted system for phenotyping Crohn's disease, categorized by: (1) Age at diagnosis (A): A1 (<16 years), A2 (17–40 years), A3 (>40 years). (2) Disease location (L): L1 (small intestine), L2 (colon), L3 (ileocolon), L4 (upper gastrointestinal). (3) Disease behavior (B): B1 (non-stricturing, non-penetrating), B2 (stricturing), B3 (penetrating). This classification helps standardize disease characterization and has been validated in numerous clinical studies [14,15].

## Data preprocessing

Missing value handling was determined based on variable characteristics and missing rates. For continuous variables with a missing rate below 10%, mean imputation was used. For categorical variables with a missing rate between 10% and 20%, mode imputation was applied and marked. Variables with a missing rate greater than 20% were evaluated using chi-square tests (for categorical variables) or t-tests (for continuous variables). If $P > 0.05$ and feature importance ranking was low, these variables were excluded. Continuous variables were standardized using the Z-score method, where new values = (original value – mean)/ standard deviation, ensuring a mean of 0 and standard deviation of 1 for model training. Categorical variables were encoded using one-hot encoding to ensure model recognition. Data were strictly screened to remove obviously incorrect or logically contradictory records, such as unreasonable extreme values of laboratory indicators [16,17].

## Baseline data statistics

The entire data analysis process was conducted using R software (version 4.3.2). For normally distributed continuous variables, characteristics were expressed as mean ± standard deviation (SD). For non-normally distributed numerical variables, the median and interquartile range (IQR) were used. For categorical variables, counts and percentages (n, %) were used. Comparisons between groups of normally distributed continuous variables were made using one-way analysis of variance (ANOVA). For non-normally distributed continuous variables, the Mann-Whitney U test (for two-group comparisons) and the Kruskal-Wallis H test (for multiple-group comparisons) were used. These non-parametric tests do not rely on distribution assumptions and can effectively compare differences between groups. For categorical variables, the chi-square test and Fisher's exact test were used to determine associations or differences between different categories. Statistical significance was set at $P < 0.01$, indicating that when the P value of the test result was less than 0.01, the difference between the studied variables or groups was considered practically significant and not due to chance.

## Model construction and training

Six machine learning models—Random Forest (RF), Gradient Boosting Machine (GBM), XGBoost, LightGBM, Cat-Boost, and AdaBoost—were selected to build predictive models. The collected data were randomly divided into an 80% training set for model training and parameter optimization, and a 20% test set for evaluating the model's generalization performance. In the training set, a 5-fold cross-validation method was further adopted to optimize the model parameters, thereby enhancing the stability and reliability of the model. RF makes decisions through voting among multiple decision trees and improves performance by adjusting the number of trees and node sampling parameters. GBM iteratively trains decision trees to fit residuals, focusing on optimizing the learning rate and tree depth. XGBoost is based on gradient boosting and enhances performance by adjusting parameters such as tree depth and learning rate during training. LightGBM uses a unique algorithm and mainly adjusts parameters such as the number of leaf nodes and learning rate. CatBoost optimizes for categorical features and adjusts parameters such as tree depth and learning rate during training. AdaBoost iteratively trains weak learners and adjusts sample weights based on error rates, optimizing the number of weak learners and learning rate. Through repeated training and validation, the parameters of each model were optimized to accurately predict the clinical remission status of CD patients after 12 weeks and 48 weeks of ADA treatment [18–20].

## Model evaluation metrics

A series of performance evaluation metrics were used to assess the effectiveness of the models, including accuracy (the proportion of correctly predicted samples out of the total samples), sensitivity (the model's ability to correctly identify positive cases), specificity (the model's ability to correctly identify negative cases), recall (similar to sensitivity), precision (the proportion of truly positive cases among those predicted as positive by the model), and the area under the curve (AUC) for binary classification models, with values closer to 1 indicating stronger discrimination ability.

## SHAP analysis method

The SHAP algorithm was applied to the trained optimal model. SHAP calculates the marginal contribution of each feature to the prediction result under all possible feature combinations to obtain the SHAP value. A positive value indicates that the feature promotes the prediction towards remission, while a negative value indicates the opposite. The SHAP-summary plot displays the ranking of feature importance, and the SHAP-force plot analyzes the influence of features on individual samples, providing an in-depth interpretation of the model's decision-making mechanism [21].

## Results

### Baseline characteristics of patients

This study included 244 adult CD patients treated with ADA at the Second Xiangya Hospital from January 2017 to April 2024. Detailed baseline characteristics were analyzed (Table 1).

Regarding gender, statistical tests showed no significant differences in the distribution of genders between the clinical remission group and the non-remission group at both 12 weeks ($P = 0.354$) and 48 weeks ($P = 0.223$). This suggests that gender may have a limited impact on the clinical remission of ADA treatment for CD.

According to the Montreal age classification, the A2 group accounted for 70.9% (61 cases) in the remission group and 63.9% (101 cases) in the non-remission group at 12 weeks ($P = 0.335$); at 48 weeks, it accounted for 70.5% (105 cases) in the remission group and 61.1% (58 cases) in the non-remission group ($P = 0.166$). These results indicate no significant differences in the distribution of different age groups between the two groups, although other factors should be considered comprehensively.

**Table 1. Comparative Analysis of Baseline Data.**

| Variable | 12W | | | 48W | | |
|---|---|---|---|---|---|---|
| | Remission Group (n = 86) | Non-Remission Group (n = 158) | p-value | Remission Group (n = 149) | Non-Remission Group(n = 95) | P |
| Male | 55 (64.0%) | 90 (57.0%) | 0.354 | 93 (62.4%) | 51 (53.7%) | 0.223 |
| Smoking | 35 (40.7%) | 63 (39.9%) | 0.991 | 60 (40.3%) | 39 (41.1%) | 0.990 |
| Montreal Age | | | | | | |
| A2 | 61 (70.9%) | 101 (63.9%) | 0.335 | 105 (70.5%) | 58 (61.1%) | 0.166 |
| A3 | 25 (29.1%) | 57 (36.1%) | 0.335 | 44 (29.5%) | 37 (38.9%) | 0.166 |
| Montreal Location | | | | | | |
| L1 | 31 (36.0%) | 49 (31.0%) | 0.511 | 46 (30.9%) | 33 (34.7%) | 0.625 |
| L2 | 20 (23.3%) | 42 (26.6%) | 0.677 | 36 (24.2%) | 26 (27.4%) | 0.682 |
| L3 | 27 (31.4%) | 64 (40.5%) | 0.205 | 59 (39.6%) | 33 (34.7%) | 0.530 |
| L4 | 2 (2.3%) | 1 (0.6%) | 0.590 | 2 (1.3%) | 1 (1.1%) | 0.692 |
| L1 + L4 | 6 (7.0%) | 2 (1.3%) | 0.044 | 6 (4.0%) | 2 (2.1%) | 0.650 |
| Montreal Behavior | | | | | | |
| B1 | 34 (39.5%) | 64 (40.5%) | 0.991 | 56 (37.6%) | 41 (43.2%) | 0.463 |
| B2 | 35 (40.7%) | 62 (39.2%) | 0.932 | 60 (40.3%) | 38 (40.0%) | 0.927 |
| B3 | 14 (16.3%) | 31 (19.6%) | 0.638 | 29 (19.5%) | 16 (16.8%) | 0.730 |
| B2 + B3 | 3 (3.5%) | 1 (0.6%) | 0.250 | 4 (2.7%) | 0 (0.0%) | 0.274 |
| BMI | 20.57 ± 3.23 | 20.27 ± 2.96 | 0.465 | 20.30 ± 3.09 | 20.49 ± 3.02 | 0.626 |
| WBC | 6.75 ± 2.16 | 6.61 ± 1.43 | 0.545 | 6.77 ± 1.92 | 6.49 ± 1.33 | 0.216 |
| Hb | 136.78 ± 25.55 | 139.90 ± 26.05 | 0.369 | 136.44 ± 25.96 | 142.51 ± 25.41 | 0.074 |
| Alb | 39.34 ± 4.91 | 39.90 ± 5.39 | 0.429 | 39.30 ± 5.55 | 40.33 ± 4.63 | 0.133 |
| Fc | 106.93 ± 137.70 | 77.62 ± 116.08 | 0.079 | 95.68 ± 125.52 | 75.83 ± 122.92 | 0.226 |
| CRP | 10.19 ± 15.59 | 12.03 ± 18.45 | 0.499 | 8.28 ± 15.60 | 10.78 ± 18.44 | 0.101 |
| plt | 311.50 (69.50) | 317.50 (70.00) | 0.311 | 320.00 (75.00) | 314.00 (67.00) | 0.260 |
| ESR | 21.50 (13.00) | 22.00 (12.00) | 0.300 | 22.00 (13.00) | 22.00 (11.00) | 0.262 |

Smoking history did not show a significant association with clinical remission. At 12 weeks, the proportion of smokers was 40.7% (35 cases) in the remission group and 39.9% (63 cases) in the non-remission group (P = 0.991); at 48 weeks, the proportions were 40.3% (60 cases) and 41.1% (39 cases), respectively (P = 0.990).

Based on the Montreal behavior classification, most types showed no significant differences in distribution between the remission and non-remission groups at both 12 weeks and 48 weeks, with only a few combinations showing trends toward differences. This suggests that the impact of disease type on treatment remission requires further investigation.

The Montreal location classification revealed that the L1 + L4 combination had a statistically significant difference in distribution between the two groups at 12 weeks (P = 0.044), while at 48 weeks, there were no statistically significant differences in any location classifications. This indicates that disease location may influence treatment remission, but the underlying mechanisms are complex and warrant further exploration.

In terms of laboratory indicators, the BMI of the remission group at 12 weeks was 20.57 ± 3.23, and that of the non-remission group was 20.27 ± 2.96 (P = 0.465); at 48 weeks, the BMI of the remission group was 20.30 ± 3.09, and that of the non-remission group was 20.49 ± 3.02 (P = 0.626). Indicators such as WBC, Hb, Alb, Fc, CRP, plt, and ESR showed varying means, standard deviations, or medians and interquartile ranges in different remission status groups. Some indicators, such as Hb at 48 weeks, showed a trend toward near-statistical significance (P = 0.074), but overall, most indicators did not reach a significant level of difference between groups. However, these indicators reflecting the patient's

nutritional, inflammatory, and immune physiological states may be interrelated and affect treatment outcomes, requiring further comprehensive analysis.

## Model performance comparison

This study evaluated the performance of six machine learning models: Random Forest (RF), Gradient Boosting Machine (GBM), XGBoost, LightGBM, CatBoost, and AdaBoost. The results are summarized in Fig 1.

When predicting clinical remission at 12 weeks post-ADA treatment (Fig 1a), the models exhibited varying performance characteristics. XGBoost and GBM achieved nearly identical accuracy rates of 0.813, demonstrating their high performance in distinguishing between remission and non-remission patients. RF, CatBoost, and LightGBM exhibited slightly lower accuracy rates at 0.803, 0.803, and 0.775, respectively, whereas AdaBoost showed the lowest accuracy rate of 0.672. In terms of precision and recall, XGBoost and GBM demonstrated superior performance with values of 0.811 and 0.813, respectively, effectively identifying patients in the true clinical remission phase. In contrast, AdaBoost exhibited relatively lower precision and recall rates at 0.664 and 0.672, respectively. The F1-scores of the models also varied significantly, with GBM and XGBoost achieving higher scores of 0.812 and 0.811, respectively, which were notably higher than those of other models. Furthermore, RF achieved an AUC value of 0.915, demonstrating its robust ability to distinguish between patients in remission and non-remission phases. GBM and XGBoost followed closely with AUC values of 0.906 and 0.891, respectively. Conversely, AdaBoost's AUC value was only 0.664, reflecting its weaker discriminatory capacity.

For predicting clinical remission at 48 weeks (Fig 1b), the model performance exhibits both similarities and differences compared to the results at 12 weeks. CatBoost achieves the highest accuracy of 0.859, while RF, XGBoost, and LightGBM attain a high accuracy of 0.822. In terms of precision, recall, and F1-score, CatBoost demonstrates the best performance with respective values of 0.860, 0.859, and 0.859, followed closely by RF, XGBoost, and LightGBM, which also exhibit strong performance. In contrast, AdaBoost and GBM show relatively weaker metrics across these categories. Regarding AUC values, RF achieves the highest score of 0.935, followed by LightGBM (0.928), CatBoost (0.926), and XGBoost (0.924). AdaBoost's AUC value is notably lower at 0.635, reflecting its limited discrimination ability.

Based on the analysis of 5 key indicators, when predicting clinical remission in CD patients treated with ADA for 12 weeks, both GBM and XGBoost models demonstrated significantly superior performance compared to other models. When predicting clinical remission in CD patients treated with ADA for 48 weeks, CatBoost, XGBoost, RF, and LightGBM models all exhibited strong performance. Overall, XGBoost consistently performed well across different datasets, suggesting its ability to provide reliable prediction results and serve as a valuable tool for clinicians in evaluating patient treatment outcomes.

## SHAP analysis results

To gain a deeper understanding of the role of each feature in the predictive model, this study employed the SHAP algorithm to analyze the trained optimal model (XGBoost). In predicting clinical remission at 12 weeks (Fig 2), the feature importance ranking (Fig 2a and Fig 2b) clearly shows significant variations in the influence of different features. Fc stands out among numerous features, with a relatively high average Shapley value, indicating a strong negative contribution to the prediction outcome. Patients with lower Fc values tend to achieve remission at 12 weeks, suggesting that Fc levels are closely associated with clinical remission currently point. This finding implies that Fc may play a crucial role in immune regulation or drug action mechanisms. CRP also holds an important position as a key indicator of inflammatory response. Its level reflects the degree of inflammatory activity in CD, thereby significantly influencing the prediction of clinical remission. Additionally, other features such as gender (Male), BMI, Alb, and Hb should not be overlooked.

In the Force Plot analysis (Fig 2c and Fig 2d), the specific impact of each feature on the prediction outcome can be visually observed. The positive or negative magnitude of each feature's Shapley value represents its promoting or inhibiting effect on the prediction outcome. Positive Shapley values drive the prediction towards clinical remission, while

## (a): 12W

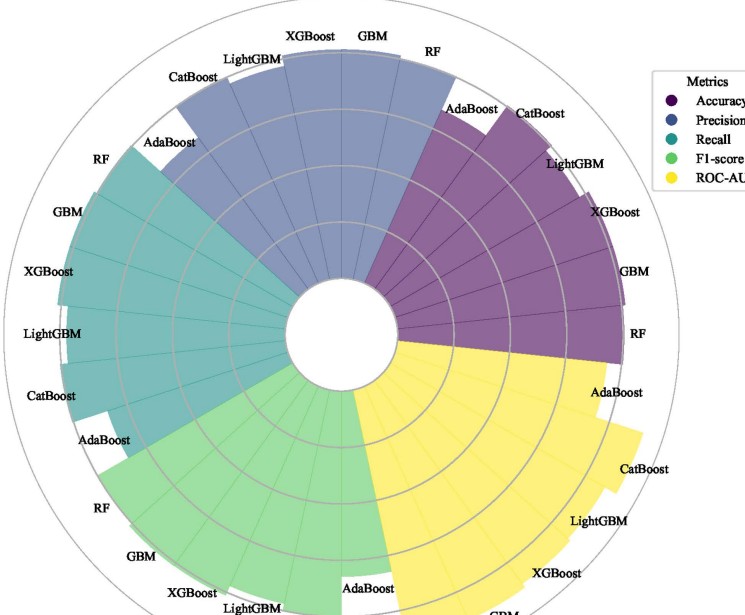

| Model | Accuracy | Precision | Recall | F1-score | ROC-AUC |
|---|---|---|---|---|---|
| RF | 0.803 | 0.802 | 0.803 | 0.801162 | 0.915 |
| GBM | 0.813 | 0.811 | 0.813 | 0.812 | 0.906 |
| XGBoost | 0.813 | 0.811 | 0.813 | 0.811 | 0.891 |
| LightGBM | 0.775 | 0.774 | 0.775 | 0.774 | 0.880 |
| CatBoost | 0.803 | 0.802 | 0.803 | 0.802 | 0.926 |
| AdaBoost | 0.672897 | 0.664 | 0.672 | 0.657 | 0.747 |

## (b): 48W

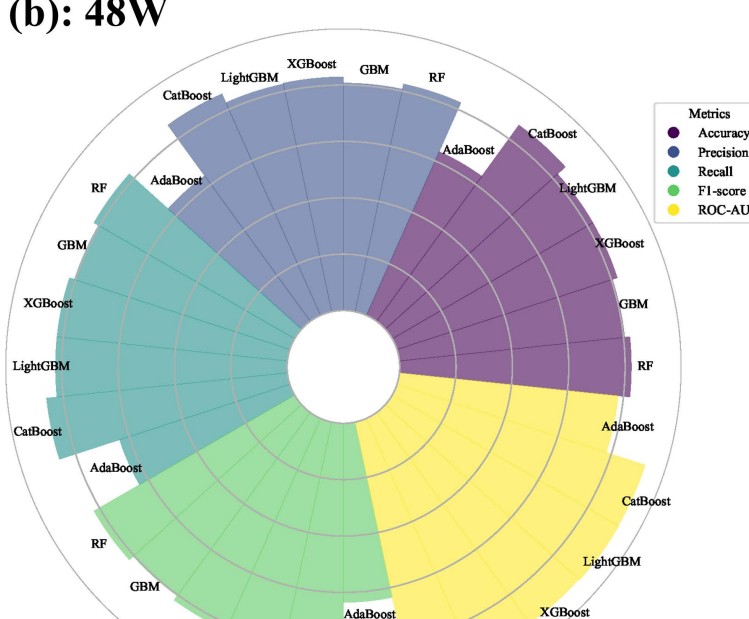

| Model | Accuracy | Precision | Recall | F1-score | ROC-AUC |
|---|---|---|---|---|---|
| RF | 0.822 | 0.825 | 0.822 | 0.822 | 0.935 |
| GBM | 0.803 | 0.806 | 0.803 | 0.803 | 0.904 |
| XGBoost | 0.822 | 0.828 | 0.822 | 0.821 | 0.924 |
| LightGBM | 0.822 | 0.825 | 0.822 | 0.822 | 0.928 |
| CatBoost | 0.859 | 0.860 | 0.859 | 0.859 | 0.926 |
| AdaBoost | 0.635 | 0.637 | 0.635 | 0.635 | 0.781 |

**Fig 1. Evaluation and Comparative Analysis of Various Machine Learning Models.** (a: The performance of six machine learning models in predicting clinical remission of CD patients treated with adalimumab for 12 weeks was evaluated. b: The performance of six machine learning models in predicting clinical remission of CD patients treated with adalimumab for 48 weeks was evaluated.).

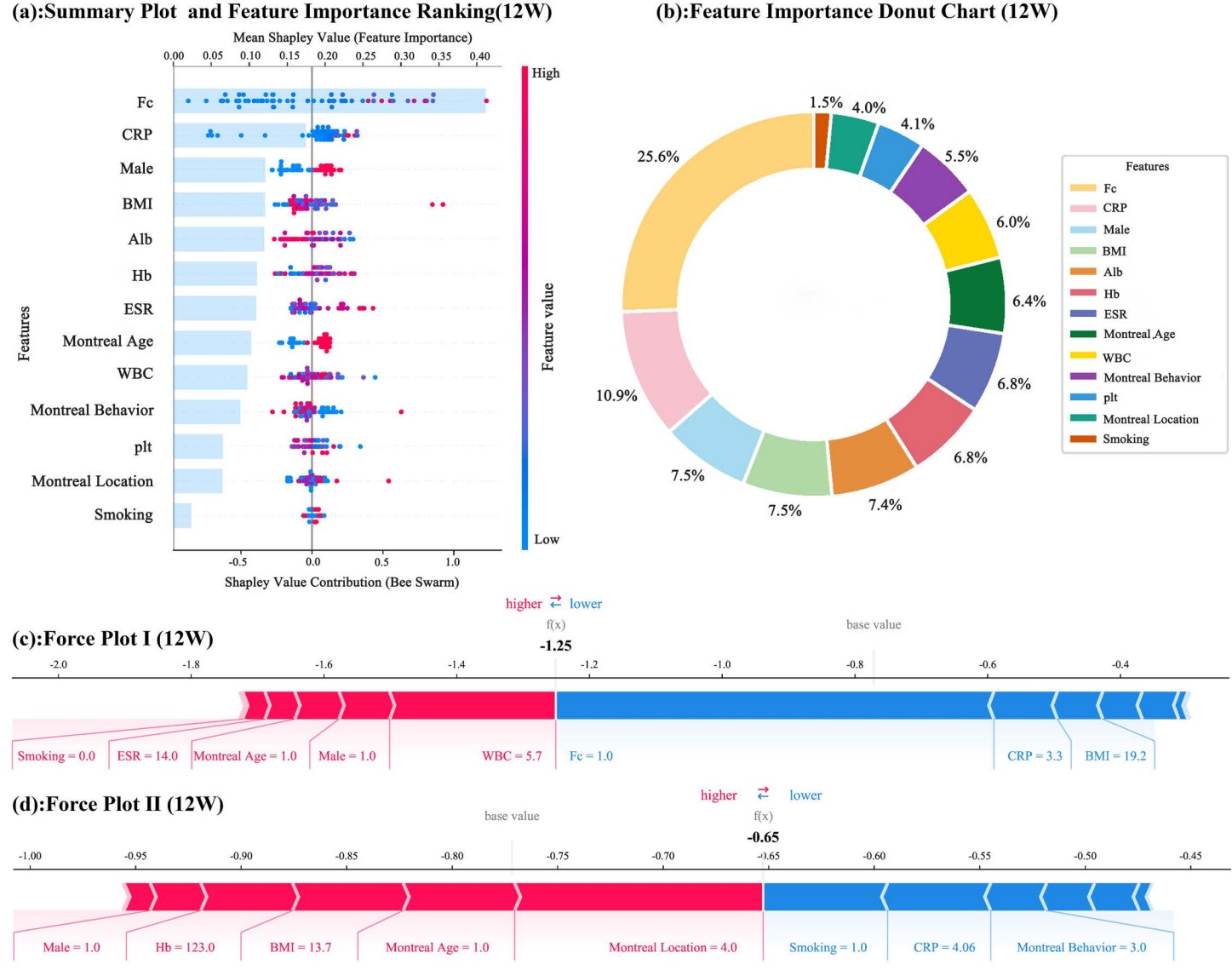

**Fig 2. SHAP Analysis of the 12 Weeks Prediction Results.** (In the Summary Plot and Feature Importance Donut Chart, features such as Fc and CRP exhibit significant influence. The average Shapley value for Fc is notably high, indicating a strong negative contribution to the prediction results. Specifically, lower Fc values are associated with a higher likelihood of achieving 12 weeks clinical remission, underscoring its critical role in immune regulation or drug action mechanisms. The Force Plot analysis provides an intuitive visualization of each feature's impact on individual sample predictions. The positive or negative sign and magnitude of the Shapley values for each feature reflect their respective promoting or inhibiting effects on clinical remission predictions. Multiple features interact synergistically to collectively determine the final prediction outcome.).

negative values have the opposite effect. From the overall impact of features, multiple features interact and jointly determine the prediction direction. In the Force Plot I sample, a combination of lower Fc and CRP values, along with other favorable features, may lead the prediction outcome to be more inclined towards clinical remission. Conversely, in the Force Plot II sample, if certain key features have unfavorable values, such as very low Hb levels or BMI in an unfavorable range, even if other features have positive effects, it may still result in a prediction outcome unfavorable for clinical remission.

In the model for predicting clinical remission at 48 weeks (Fig 3), the ranking of feature importance changed (Fig 3a and Fig 3b). Fc remained a significant influencing factor, but the importance of Hb markedly increased, with a higher average Shapley value. This increase in Hb's importance may be attributed to its growing impact on treatment outcomes as the disease progresses. Features such as ESR, plt, and CRP also played crucial roles in the prediction. ESR continuously reflects the degree of inflammation, while plt is associated with physiological processes like immunity and coagulation. These factors collectively influence the prediction of clinical remission at 48 weeks.

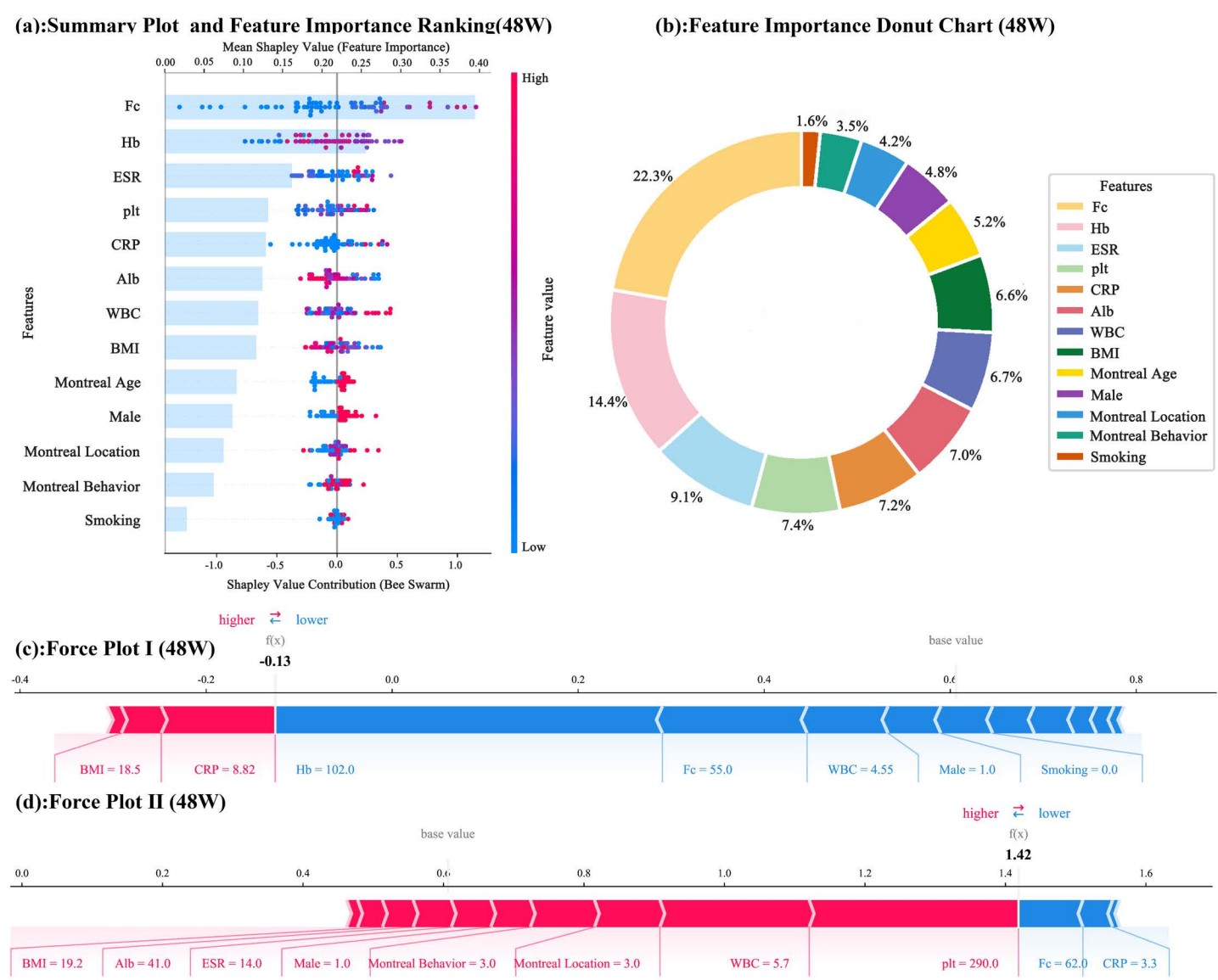

**Fig 3. SHAP Analysis of the 48 Weeks Prediction Results.** (In the Summary Plot and Feature Importance Donut Chart, Fc remains a significant factor. Hemoglobin (Hb) has become increasingly important, with its average Shapley value rising, likely due to anemia's growing impact on treatment efficacy as the disease progresses. Erythrocyte sedimentation rate (ESR), platelets (plt), and C-reactive protein (CRP) also play crucial roles in predicting 48 weeks clinical remission. ESR reflects inflammation levels, while plt influences immune and coagulation processes. The Force Plot analysis visually demonstrates how different feature values combine to affect predictions. Changes in key features like Hb and ESR can shift prediction results toward or away from clinical remission.).

In the corresponding Force Plot analysis (Fig 3c and Fig 3d), the combined effects of each feature across all samples can be observed. Different samples exhibit varying values for each feature, leading to diverse prediction results. In some samples, if the Hb value is high and the ESR value is low, along with other features being within favorable ranges for disease remission, the prediction result is more likely to lean towards clinical remission. Conversely, if key features have unfavorable values, such as low Hb levels and high CRP levels, even if other features have positive influences, the prediction result may still be unfavorable for clinical remission.

Combining the SHAP analysis results at 12 weeks and 48 weeks, it is evident that certain features, such as Fc and CRP, consistently influence the prediction results across different time points, demonstrating their stability in assessing clinical remission. However, the importance of other features, like Hb, varies over time, indicating that the key factors influencing clinical remission differ at various stages of the disease process. These findings provide a basis for a deeper understanding of the mechanisms underlying disease treatment responses and can assist clinicians in focusing on key indicators based on the disease stage to develop more effective personalized treatment plans.

## Discussion

### Interpretation of feature importance

In this study, the XGBoost model demonstrated outstanding performance. This model efficiently integrates multiple weak learners and accurately captures the complex interactions among features, thereby excelling in predicting clinical remission in CD patients treated with ADA. Although many variables did not exhibit significant univariate associations (Table 1), they were retained in the machine learning model to uncover potential interactions with other features. Unlike traditional statistical methods that primarily focus on linear relationships, machine learning can explore complex nonlinear relationships between variables, thereby enhancing its value in integrating multi-dimensional clinical data.

Based on this, SHAP analysis was employed to identify key features that significantly influence clinical remission. These features are closely related to inflammatory responses, nutritional metabolism, and disease location, aligning well with the pathological and physiological mechanisms of CD.

The study found that Fc and CRP were crucial for prediction results at different time points. There was a strong correlation between Fc levels and clinical remission at 12 weeks, with lower Fc values often indicating a higher likelihood of remission. This suggests that Fc may play a central role in immune regulation or the mechanism of action of ADA. Fc is secreted by neutrophils and directly reflects the degree of intestinal mucosal inflammation. A low Fc value may indicate mucosal healing, which, combined with ADA's local anti-inflammatory effect in inhibiting TNF-α, promotes remission [22,23]. Additionally, CRP, as an acute-phase protein, indicates systemic inflammation. Elevated CRP levels suggest uncontrolled inflammation [24]. Studies have shown [22] that persistent inflammation can disrupt intestinal homeostasis, causing mucosal damage, abnormal immune cell activation, and cytokine network imbalance. These changes can interfere with drug-target binding and hinder drug efficacy, thereby impeding clinical remission, consistent with the significant role of CRP observed in this study.

As the disease progresses, Hb's importance significantly increases at 48 weeks. In the early stages, compensatory mechanisms may maintain bodily functions, but prolonged treatment exacerbates the impact of anemia on physical condition and treatment tolerance [25]. Anemia weakens the body, affecting oxygen transport and supply, leading to hypoxia in tissues and organs, which weakens the immune system and reduces treatment responsiveness [26]. From a cellular metabolism perspective, hypoxia affects cell proliferation, differentiation, and repair, hindering mucosal healing and disfavoring clinical remission [27]. Literature [28,29] also indicates that anemia lowers quality of life and increases infection risks, collectively highlighting Hb's increasing significance in predicting clinical remission at 48 weeks, consistent with our findings.

Indicators reflecting patient nutritional status, such as BMI and Alb, also play significant roles in the prediction model. BMI measures body fatness and health status, with its reasonable range reflecting balanced nutritional intake and consumption. An appropriate BMI signifies sufficient energy reserves, supporting immune function [30]. Alb, synthesized by the liver, reflects liver synthetic function and overall nutritional status. Normal Alb levels indicate adequate nutritional reserves and good liver function, maintaining intestinal barrier integrity and enhancing response to biological treatments. Studies show [31] that proper nutritional support, including diets rich in proteins, vitamins, and trace elements, and necessary enteral or parenteral nutrition, can improve immune and intestinal barrier functions, enhance treatment sensitivity and underscore the importance of BMI and Alb in the predictive model.

Overall, the SHAP analysis in the results section revealed dynamic changes in feature importance at 12 and 48 weeks. Both Fc and CRP were identified as key influencing factors at both time points. However, Fc exhibited a stronger negative contribution to clinical remission at 12 weeks, with its average SHAP value contribution decreasing by 48 weeks, suggesting that the mucosal healing effect of adalimumab may stabilize over time. The importance of Hb significantly increased at 48 weeks, aligning with the cumulative impact of anemia on treatment tolerance during disease progression. Additionally, ESR demonstrated a higher average SHAP value at 48 weeks compared to 12 weeks, indicating that while CRP serves as the primary predictor in the early stages, ESR plays a more critical role in long-term remission. Contributions from BMI and Alb remained relatively stable, underscoring the sustained influence of nutritional status on treatment efficacy. These temporal variations highlight the necessity of prioritizing intestinal inflammation markers (Fc, CRP) during early treatment monitoring and focusing on anemia (Hb) and chronic inflammation (ESR) in long-term management, thereby providing a robust foundation for a staged clinical strategy in CD.

## Research significance and prospects

This study systematically evaluates six machine learning models to identify the optimal predictor for adalimumab response, incorporating SHAP analysis at both 12 and 48 weeks to uncover temporal shifts in feature importance. By integrating multi-dimensional clinical data, including demographic characteristics, laboratory indicators, and disease phenotypes, this model not only facilitates dynamic predictions of clinical remission at 12- and 48-weeks post-ADA treatment but also elucidates the time-dependent characteristics of key influencing factors via SHAP explainability analysis. Its significance is reflected in the following aspects:

The model quantifies the interactions among multiple factors, providing clinicians with a reliable tool for predicting treatment responses. For instance, patients with lower baseline levels of Fc and CRP exhibit a significantly higher probability of achieving clinical remission at 12 weeks, suggesting that standardized ADA treatment could be prioritized for these individuals. Conversely, for patients predicted to have a poor response, early adjustments to the treatment strategy may be warranted.

Additionally, the temporal evolution of feature importance revealed by SHAP analysis offers distinct monitoring targets throughout the entire treatment process for Crohn's disease (CD) management. In the early stages of treatment, priority should be given to monitoring local intestinal inflammation markers and systemic inflammatory burden. As treatment advances, focus should gradually shift toward managing anemia and nutritional status. This "stage-specific" monitoring strategy enables the transition from empirical decision-making to a data-driven, precision-oriented approach in clinical management.

## Research limitations

Despite achieving significant results, this study has limitations. First, it used a single-center sample with a limited size, potentially introducing selection bias. Patients from one hospital may not fully represent all CD patients, limiting the model's generalizability and applicability in diverse medical settings.

While the XGBoost model exhibits strong discriminative performance, it is worth emphasizing that this study prioritizes discriminative accuracy over probabilistic calibration or clinical decision thresholds. Future studies utilizing multi-center data could further assess the model's clinical utility through decision curve analysis or calibration evaluations, especially in varied treatment contexts.

Additionally, emerging multi-omics technologies, such as gut microbiomics and metabolomics, offer new insights into disease pathogenesis and treatment response [32]. The occurrence and development of CD are closely linked to imbalances in the intestinal microbiota and metabolic products [33]. Our model, based solely on clinical and laboratory data, lacks comprehensive omics data, potentially missing critical factors influencing treatment outcomes.

To address these limitations, future research should conduct multi-center collaborative studies to expand sample sizes and diversity, reducing selection bias and improving generalizability. Standardized data collection protocols should be established to enhance data quality control. Integrating multi-omics data will allow for a more comprehensive and precise predictive model, identifying additional biomarkers related to treatment response and further improving model performance.

## Supporting information

**S1 File. The original source code and dataset.**
(ZIP)

## Author contributions

**Conceptualization:** Feihong Deng.

**Data curation:** Pianpian Xia.

**Formal analysis:** Pianpian Xia.

**Funding acquisition:** Feihong Deng.

**Investigation:** Pianpian Xia.

**Methodology:** Feihong Deng, Deliang Liu.

**Software:** Pianpian Xia.

**Supervision:** Deliang Liu.

**Visualization:** Pianpian Xia.

**Writing – original draft:** Pianpian Xia.

**Writing – review & editing:** Feihong Deng.

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
