## [Decision Letter · Decision Letter 0]

6 May 2025

Dear Dr. Deng,

We look forward to receiving your revised manuscript.

Kind regards,

Ahmed Qasim Mohammed Alhatemi, Mbchb

Academic Editor

PLOS ONE

 [Funding This work was supported by the National Natural Science Foundation of China (Grant number 82470586), and the grantee is Dr. Feihong Deng.].

6. "Please include a copy of Table 1 which you refer to in your text on page 6.

Additional Editor Comments (if provided):

Reviewers' comments:

Reviewer's Responses to Questions

**Comments to the Author**

1. Is the manuscript technically sound, and do the data support the conclusions?

Reviewer #1: Yes

Reviewer #2: Yes

2. Has the statistical analysis been performed appropriately and rigorously?

Reviewer #1: Yes

Reviewer #2: Yes

3. Have the authors made all data underlying the findings in their manuscript fully available?

Reviewer #1: No

Reviewer #2: Yes

4. Is the manuscript presented in an intelligible fashion and written in standard English?

Reviewer #1: Yes

Reviewer #2: Yes

Reviewer #1: Overall Assessment

- The paper is well-written and attempts to address an important topic in the Crohn's disease literature.

- The authors apply multiple ML models with a standard 80/20 split in a very standard way.

- The evaluation metrics used in the study are standard.

- Ethics statement and funding are mentioned explicitly.

- However, certain areas require revisions.

Major Comments

- There is no dedicated 'Literature Review' section and hence no way to know where the study can be placed within the existing literature and the novelty it brings.

- The study mentions various methods including ML models, SHAP and data preprocessing methods without any citations.

- There is no description or citation of 'Montreal classification' which makes it difficult for an interdisciplinary reader to understand.

- Manuscript is not formatted according to PLOS ONE guidelines.

Minor Comments

- Citations 13 and 16 refer to the same study. This redundancy must be corrected.

- Association between BMI & clinical remission and definitions of Alb and Hb (lines 565-569) are uncited.

- PLOS ONE requires making the full data available without restriction. However, if ethical issues arise in doing so, please refer to the guidelines.

Reviewer #2: General Assessment

This manuscript developed and validated machine learning-based models to predict clinical remission in Crohn’s disease patients receiving Adalimumab therapy, addressing a clinically significant topic with a generally appropriate study design. However, substantial concerns remain regarding external validation, feature selection methodology, statistical rigor, and translational interpretation. The robustness and generalizability of the conclusions require further substantiation. Major revision is recommended before further consideration.

Comments and Suggestions for Authors

1. Descriptions of hyperparameter search ranges, cross-validation strategies, and overfitting prevention measures (e.g., regularization, early stopping) are insufficient. A standardized and detailed account of the model training process is necessary to ensure methodological transparency and reproducibility.

2. The statistical analysis is limited to basic reporting of AUC values without clarification on multiple hypothesis testing corrections (e.g., DeLong test). Additionally, important performance metrics such as sensitivity, specificity, PPV, and NPV are insufficiently reported. A more comprehensive and standardized statistical analysis is recommended.

3. It is suggested that the authors expand the discussion section slightly by briefly comparing their findings with existing studies on machine learning models for anti-TNF response prediction (e.g., Infliximab-related models), thereby highlighting the novelty and improvements offered by this study.

4. The current description of missing data imputation (mean or mode) is brief and lacks a discussion of missingness mechanisms (MAR or MNAR). It is suggested to justify the chosen imputation approach and briefly assess its potential impact on the study conclusions.

5. Currently, SHAP analyses at 12 and 48 weeks are reported separately. It is recommended to add a comparative discussion of feature importance dynamics between these two time points to better understand the evolving key factors during disease progression.

**Do you want your identity to be public for this peer review?** For information about this choice, including consent withdrawal, please see our Privacy Policy

Reviewer #1: **Yes: ** Hasin Jawad Ali

Reviewer #2: No

---

## [Author Response · Author response to Decision Letter 1]

30 Jun 2025

All responses to the journal's requirements and the comments of the two reviewers are detailed in the attached document

---

## [Decision Letter · Decision Letter 1]

30 Jul 2025

Dear Dr. Deng,

Thank you for submitting your manuscript to PLOS ONE. After careful consideration, we feel that it has merit but does not fully meet PLOS ONE’s publication criteria as it currently stands. Therefore, we invite you to submit a revised version of the manuscript that addresses the points raised during the review process.

We look forward to receiving your revised manuscript.

Kind regards,

Ahmed Qasim Mohammed Alhatemi, Mbchb

Academic Editor

PLOS ONE

Journal Requirements:

Reviewers' comments:

Reviewer's Responses to Questions

**Comments to the Author**

Reviewer #1: (No Response)

Reviewer #2: All comments have been addressed

2. Is the manuscript technically sound, and do the data support the conclusions?

Reviewer #1: Yes

Reviewer #2: Yes

3. Has the statistical analysis been performed appropriately and rigorously?

Reviewer #1: Yes

Reviewer #2: Yes

4. Have the authors made all data underlying the findings in their manuscript fully available?

Reviewer #1: No

Reviewer #2: Yes

5. Is the manuscript presented in an intelligible fashion and written in standard English?

Reviewer #1: Yes

Reviewer #2: Yes

Reviewer #1: The authors have done a very good job at addressing the concerns raised earlier. Most major and minor issues have been resolved.. However, I would still like to recommend a few revisions:

- The section "Research Significance and Prospects" at the very end appears to be a form of "Related Work" section. I believe it would be better to include it after the "Introduction" section and make it more comprehensive.

- In "Baseline Data Statistics" subsection, there is no rationale or citation for selecting statistical significance at less than 0.01. An explanation would enrich it.

- The dataset is now available upon request. But Plos One requires the dataset to be fully public.

Reviewer #2: The authors have thoroughly addressed previous comments. The revised manuscript is clear, methodologically sound, and clinically relevant. I recommend acceptance for publication.

**Do you want your identity to be public for this peer review?** For information about this choice, including consent withdrawal, please see our Privacy Policy

Reviewer #1: No

Reviewer #2: No

---

## [Author Response · Author response to Decision Letter 2]

11 Aug 2025

Please review the reply comments document.

---

## [Editor Report · Decision Letter 2]

17 Aug 2025

Development and Validation of a Machine Learning-Based Predictive Model for Clinical Remission in Crohn's Disease Patients Receiving Adalimumab Therapy

PONE-D-25-14568R2

Dear Dr. Deng,

We’re pleased to inform you that your manuscript has been judged scientifically suitable for publication and will be formally accepted for publication once it meets all outstanding technical requirements.

Kind regards,

Ahmed Qasim Mohammed Alhatemi, Mbchb

Academic Editor

PLOS ONE
---

## [Editor Report · Acceptance letter]

PONE-D-25-14568R2

PLOS ONE

Dear Dr. Deng,

I'm pleased to inform you that your manuscript has been deemed suitable for publication in PLOS ONE. Congratulations! Your manuscript is now being handed over to our production team.

Kind regards,

on behalf of

Dr. Ahmed Qasim Mohammed Alhatemi

Academic Editor

PLOS ONE